# Self-Assembly of Supramolecular Architectures by the Effect of Amino Acid Residues of Quaternary Ammonium Pillar[5]arenes

**DOI:** 10.3390/ijms21197206

**Published:** 2020-09-29

**Authors:** Anastasia Nazarova, Dmitriy Shurpik, Pavel Padnya, Timur Mukhametzyanov, Peter Cragg, Ivan Stoikov

**Affiliations:** 1A.M.Butlerov Chemical Institute, Kazan Federal University, Kremlevskaya, 18, 420008 Kazan, Russia; anas7tasia@gmail.com (A.N.); dnshurpik@mail.ru (D.S.); padnya.ksu@gmail.com (P.P.); timmie.m@gmail.com (T.M.); 2School of Pharmacy and Biomolecular Sciences, University of Brighton, Huxley Building, Moulsecoomb, Brighton, East Sussex BN2 4GJ, UK; p.j.cragg@brighton.ac.uk

**Keywords:** aggregation, (1S)-(+)-10-camphorsulfonic acid, host-guest systems, methyl orange dye, amino acid, pillar[5]arene

## Abstract

Novel water-soluble multifunctional pillar[5]arenes containing amide-ammonium-amino acid moiety were synthesized. The compounds demonstrated a superior ability to bind (1S)-(+)-10-camphorsulfonic acid (*S*-CSA) and methyl orange dye depending on the nature of the substituent, resulting in the formation one-to-one complexes with both guests. The formation of host-guest complexes was confirmed by ultraviolet (UV), circular dichroism (CD) and ^1^H NMR spectroscopy. This work demonstrates the first case of using *S*-CSA as a chiral template for the non-covalent self-assembly of architectures based on pillar[5]arenes. It was shown that pillar[5]arenes with glycine or L-alanine fragments formed aggregates with average hydrodynamic diameters (d) of 165 and 238 nm, respectively. It was established that the addition of *S*-CSA to the L-alanine-containing derivative led to the formation of micron-sized aggregates with d of 713 nm. This study may advance the design novel stereoselective catalysts and transmembrane amino acid channels.

## 1. Introduction

Chirality is a key factor in the action principles of biologically active macromolecules as well as in the separation processes of enantiomers reaction mixtures [1,2,3]. In view of this, considerable attention has been paid to the recognition of optically active compounds. Interest in designing enantiomerically pure compounds is due to requirements in the pharmaceutical industry, applications as sensors for asymmetric biomolecules and in creating a new generation of catalysts [4,5].

Chiral receptors based on biomacromolecules and related biomimetic structures have been widely used in recent decades [6,7,8]. In the literature on biomimetic structures capable of recognizing target analytes, synthetic analogues of peptides and proteins, which form a group of chemically and biologically related “platforms”, are the most convenient to employ. However, systems based on macrocyclic compounds are increasingly used to develop a new generation of peptidomimetics [9,10,11]. Macrocyclic structures offer great possibilities for geometry optimization of binding sites in the receptor in order to achieve complementarity between the “host” and the “guest”. The combination of interaction centers with various properties in macrocyclic systems can lead to the creation of unique three-dimensional conformationally mobile structures and the realization of the enzymatic principle of induced fit.

Molecular recognition in water is the basis for many vital processes [12,13]. A significant number of macrocyclic compounds which are able to bind guest molecules in water (cyclodextrins [14], cucurbiturils [15], calixarenes [16] and thiacalixarenes [17,18,19]) have been synthesized. Pillararenes are new type of synthetic hosts with an excellent ability to molecular recognition [20,21,22,23]. Introduction of chiral amino acid fragments into the macrocycle’s structure makes it possible to create novel water-soluble receptors based on them. A number of pillararenes containing alanine (Ala) [24,25,26], phenylalanine (Phe) [27], arginine (Arg) [28], tryptophan (Trp) [29], glycine (Gly) [30,31] and diglycine (Gly-Gly) [32] fragments have been reported in the literature. These compounds have been applied in various fields: from the binding of alkali metal cations and dyes to targeted drug delivery and use as ion channels. In the present study, an original approach to the synthesis of a new generation of peptidomimetics based on decasubstituted water-soluble pillar[5]arenes containing amide-ammonium-amino acid moiety was implemented. Introduction of these functional groups made it possible to “freeze” the macrocycles’ conformation to improve its selectivity. The binding ability of these macrocycles with a number of optically active and achiral guests containing a sulfo group was also studied. This work demonstrates the first example of (1S)-(+)-10-camphorsulfonic acid (S-CSA) being used as chiral template for non-covalent self-assembly of chiral architectures based on pillar[5]arenes.

## 2. Results and Discussion

### 2.1. Synthesis of Water-Soluble Pillar[5]arenes Containing Amino Acid Residues

Since 2008, a new class of supramolecular hosts—pillar[n]arenes—have become the object of active study, initiated by the Ogoshi group [33]. This class of synthetically accessible *p*-cyclophanes was obtained in high yield by the condensation of 1,4-dimethoxybenzene and paraformaldehyde. Pillar[n]arenes play a great role in contemporary research because they are used as building blocks to create supramolecular polymers and sensors that are sensitive to a wide range of substrates [34,35].

Amino acids are the main building blocks of many enzymes, proteins and peptides, and play important role in metabolic processes. Consequently, functionalization of macrocyclic compounds by amino acid fragments allows the creation of derivatives with antimicrobial activity, use them as artificial ion channels, for targeted drug delivery, or as cholinesterase inhibitors [24,25,26,27,28,29,30,31,32,36,37]. In this regard, the design of synthetic macrocyclic receptors containing amino acid residues is fascinating task for modern organic and supramolecular chemistry. In order to obtain quaternary ammonium salts based on pillar[5]arene containing amino acid residues the highly reactive reagents **3**, **5**, **7**, **9** (Scheme 1) were prepared according to the literature [38].

The choice of compounds **3**, **5** and **7** was determined by the presence of residues of glycine as the simplest non-chiral amino acid, and L-alanine, as the closest chiral analog of glycine, in their structure. The possibility of associates forming by ammonium derivatives of pillar[5]arenes not only through hydrogen bonding between peptide fragments, but also through π-stacking and hydrophobic effects, was the key basis for the inclusion of the *L*-phenylalanine residue in the structure of the compound **9**.

The next stage of the work was the synthesis of decasubstituted pillar[5]arene 10 containing ester fragments according to the literature [39]. Compound **11** was obtained in 72% yield (Scheme 2) by the aminolysis of pillar[5]arene 10 by 3-dimethylaminopropylamine [40].

Further interactions between macrocycle **11** containing methyl fragments at the tertiary nitrogen atom and reagents **3**, **5**, **7** and **9** were studied (Scheme 2). The mixtures was refluxed in acetonitrile for 12–14 h. Pillar[5]arenes **12**–**15** were obtained in high 87–93% yields.

Structures and compositions of these compounds were confirmed (Appendix A) by a number of physical methods (^1^H and ^13^C NMR, IR spectroscopy, mass spectrometry and elemental analysis). Purity was confirmed by melting point determination and TLC.

Jung et al. previously showed [24] that in the case of decasubstituted pillar[5]arene with L-Ala fragments, the positive Cotton effect decreases in the solvent series: water, dimethylsulfoxide (DMSO), dimethylformamide (DMF), acetonitrile. The greatest Cotton effect is observed for water, while it is practically non-existent for the solution of pillararenes in acetonitrile. The authors account for it by the enantiomeric purity of the macrocycle since the Cotton effect diminish with the decrease of the enantiomeric purity.

In this regard, we decided to study behavior of pillararenes **12**–**15** in water and DMSO. The analysis of ^1^H NMR spectra of compound **13** recorded in two different solvents (Figure 1b,c) showed that the replacement of DMSO with water led to an insignificant 0.1 ppm downfield shift of the methylene protons H^6^ of the propylene fragment.

The methyl proton H^14^ signals of the ethoxy groups in the analyzed sample are shifted downfield by 0.19 ppm. The H^13^ protons are diastereotopic therefore we can observe two superimposed triplets against each other in the NMR spectrum in water. The H^12^ protons of the methyl group shifted upfield by almost 0.54 ppm. All these changes are associated with the formation of intramolecular hydrogen bonds. This results in protonation of the carbonyl group of the alanine fragment, on which partial positive charge appears, and deprotonation of the nitrogen atom, on which a partial negative charge appears.

Cross-peaks between H^10^ protons of amide group of alanine fragment and methylene protons H^5^ and H^6^ of the propyl fragment are observed in the two-dimensional ^1^H-^1^H NOESY NMR spectrum of pillar[5]arene **13** in Figure 1. There are additional cross-peaks between H^10^ protons and H^8^ protons of the methyl groups on the quaternarized nitrogen atom. It is also worth noting the presence of cross-peaks between spatially close H^4^ protons of the amide group and H^8^, H^11^ protons of the methyl groups and methyne fragment, respectively.

In addition, correlations between the H^1^ aromatic protons and the H^5^ and H^6^ methylene protons of the propyl fragment are observed in the spectrum, as well as with H^8^ methyl protons at the quaternarized nitrogen atom. Thus, we assume that the amino acid fragments of macrocycle **13** (Figure 1a) are turned away from the macrocyclic cavity, based on analysis of its two-dimensional ^1^H-^1^H NOESY NMR, due to the electrostatic repulsion of positively charged ammonium groups.

### 2.2. Synthesis of Monomeric Analogues

Monomeric analogues **17** and **18** with L-Ala and L-Phe fragments were synthesized to further evaluate the macrocyclic platform effect as well as the effect assessment of amino acid and ammonium fragments of the models on complexing and aggregation properties, and to assess rotational changes of polarized light (Scheme 3). In this case, the synthesis of disubstituted hydroquinone does not make sense. Preparation of such compounds will result to an optically inactive meso form of hydroquinone derivative, which will not allow assessing the effect of chiral fragment on the above properties.

The reaction was carried out in boiling acetonitrile for 8 h. Compounds **17** and **18** were obtained in 95% and 94% yields respectively. The structures and compositions of the products were analyzed by ^1^H and ^13^C NMR, IR spectroscopy, mass spectrometry and elemental analysis (Appendix A).

### 2.3. CD Spectroscopy

The next stage of the work was the study of pillar[5]arenes **12**–**15** and their monomeric analogues (compounds **17** and **18**) by electron absorption and circular dichroism spectroscopy (Appendix A) in two different solvents. The absorption maximum in the UV spectra for the compounds **12**–**15** solutions is observed at 296 nm and corresponds to the π→π* transition of the pillararenes’ aryl fragments. It was shown that the Cotton effect, as expected, was not observed for pillararenes **12** and **14**, which indicated the formation of racemic mixtures (Figure 2).

Macrocycles **13** and **15** have asymmetric carbon atoms in their alanine and phenylalanine fragments. The reaction can proceed stereospecifically with predominance of one of the macrocyclic stereoisomers (either pS- or pR-), when pillar[5]arene **11** is modified by reagents with chiral centers. Thus, the Cotton effect observed in the CD spectra in the absorption region corresponding to that of the aromatic fragments indicated that **13** and **15** do not exist as racemic mixtures, but as mixtures of two stereoisomers in which one isomer prevails. It should be noted that, both the sign of the Cotton effect changes and its value decreases for macrocycles **13** and **15**, when water is replaced by DMSO. Such changes are probably related to the nature of the solvent, which stabilizes one conformational stereoisomer over the other [24]. At the same time, the different signs of the Cotton effect for pillar[5]arenes **13** and **15** containing L-amino acid fragments are owing to formation of different types of aggregates by macrocycles **13** and **15** [41].

Then the derivatives **17** and **18** were studied by CD spectroscopy (Figure 3). It was shown that the Cotton effect is shifted to the short-wave region in the case of monomers **17** and **18** in contrast to pillar[5]arenes **13** and **15** where it is observed at 300 nm. It is associated with the influence of the macrocyclic platform.

The results can be related to the aggregation processes, which were further studied. It should be noted that pillar[5]arene amino acid derivatives **12**–**15** have several binding centers: proton-donating NH groups, carbonyl oxygen lone pairs on the peptide and amide fragments, and the positively charged quaternary nitrogen atoms. Thus, these macrocycles can aggregate and bind guest molecules due to both intra- and intermolecular hydrogen bonds and ion-dipole interactions.

Aggregation of these pillar[5]arenes was studied by dynamic light scattering (DLS) in water and DMSO from 1 × 10^−5^ to 1 × 10^−3^ M. It was established that compounds **12**–**15** do not form aggregates in DMSO over the concentration range. Meanwhile, particles forming with an average hydrodynamic diameter 166 ± 2 and 237 ± 3 nm were observed (Appendix A) in water for solutions of macrocycles **12** (1 × 10^−4^ M) and **13** (1 × 10^−3^ M).

Electron microscopy allows to determine the size and shape of the aggregates. Confirmation of the formation of supramolecular aggregates by pillar[5]arenes **12** and **13** was given by TEM. It was found that compound **12** formed particles of different shapes in water (Appendix A). Formation of branched structures consisting from spherical particles of nanometer dimensions was shown for macrocycle **13** (Appendix A). This was probably due to the presence of minimum substituent (Me) in the amino acid residue turned out to be a determinative factor in the formation of branched structures. This is in good agreement with the data obtained by the dynamic light scattering, from which the average hydrodynamic diameter was 200 nm.

### 2.4. Complexation and Aggregation of Water-Soluble Pillar[5]arenes Studied by UV-, NMR Spectroscopy and Dynamic Light Scattering

To date, there are a number of works in the literature dedicated to binding substrates containing the sulfo group by pillar[5]arenes with charged ammonium fragments [40,42,43,44]. Inclusion complexes are formed with a number of aromatic sulfo derivatives, e.g., *p*-toluenesulfonic acid, methyl orange dye, etc. In view of this, the effect of introducing amino acid fragments into the structure of pillar[5]arenes was of interest on its ability to bind both chiral and non-chiral substrates containing sulfo groups.

#### 2.4.1. UV Spectroscopy

Thus, the next stage of research work was the study of macrocycles’ complexing properties by UV-spectroscopy, DLS and one- and two-dimensional NMR spectroscopy. *p*-Toluenesulfonic acid **G1**, (1S)-(+)-10-camphorsulfonic acid **G2**, camphor **G3** and methyl orange dye **G4** were chosen as substrates. Initially, the interaction between pillararenes **12**–**15** and substrates was studied by UV-spectroscopy (Appendix A). The study was conducted in water and DMSO. The solutions of **G1**–**G3** guests were added to the solutions of compounds **12**–**15** (3 × 10^−5^ M) in 1:10 ratio to study the complexation of pillar[5]arenes with guests in DMSO. A 1:1 ratio was chosen in the case of **G4** due to its high extinction coefficient. If there was no interaction between the components in the mixture, its optical density would correspond to the sum of the individual components taken in the same concentration. However, a deviation (ΔA) of the optical density of A_complex_ from that of the additive spectrum, ΣA_mixture_, where ΔA = A_complex_ − ΣA_mixture_, was found for mixtures of **13** with **G2** and **15** with **G4**. This deviation testifies an interaction between the macrocycles and those specific guest molecules. Complexation of **13** with (1S)-(+)-10-camphorsulfonic acid leads to a hyperchromic effect at 300 nm (Figure 4a). A hypochromic effect and hypsochromic shift at 500 nm are observed for complex of **15** with **G4** (Figure 4b).

The stability constants and stoichiometry of the resulting complexes were determined to quantify the complexing ability of compounds **13** and **15** with guests **G2** and **G4**. The stoichiometry of the host-guest complexes was determined by the isomolar series method. It was found that 1:1 complexes were formed in both cases. Spectrophotometric titration methods were used to determine the binding constants. The absorption spectra of the pillararene/guest systems were recorded with the concentration of pillar[5]arene remaining constant, and the ratio of camphorsulfonic acid increased from 1:1 to 1:10 for the pillar[5]arene **13**/guest **G2** system. The concentration of methyl orange dye remained constant, while pillararene ratio increased from 0.3:1 to 2:1 for the pillar[5]arene **15**/guest **G4** system (Figure 5).

The data obtained were processed by BindFit [45]; association constants of complexes with a 1:1 composition were also calculated (Appendix A). The association constant of pillar[5]arene **15** with **G4** was 4523 M^−1^ and that of macrocycle **13** with **G2** was 613 M^−1^. Moreover, the stoichiometry of the complex was confirmed by titration data processed using host-guest ratios of 1:2 and 2:1. However, in this case, the association constants of the complexes were determined with a large error.

It is obviously, that the main driving force in the formation of complexes between pillar[5]arenes **13** and **15** and **G2** and **G4** respectively is electrostatic interaction. The complex formation also becomes possible due to hydrogen bonds and hydrophobic interactions. The macrocyclic cavity meanwhile in the case **13** and **G2** practically does not participate in the complex formation according to experimental data (^1^H NMR spectra). However, according to ^1^H-^1^H NOESY NMR spectroscopy and semi-empirical PM6 method, the substituents of pillararene **13** form a pseudocavity, which forms an inclusion complex with **G2**.

#### 2.4.2. NMR Study of pillar[5]arene Complexes

One-dimensional NMR spectroscopy is a convenient way to study complexation. In this regard the next stage of investigation was study of pillar[5]arene complexes **13**-**G2** and **15**-**G4** by NMR spectroscopy (Appendix A). Solutions of compounds **13** and **15** were prepared at 1 × 10^−2^ M. Complexation with **G2** and **G4** guests (1 × 10^−2^ M) was studied at a 1:1 ratio. Unfortunately, it turned out that pillar[5]arene **15** had limited solubility at that concentration in water. Therefore, it was impossible to analyze the ^1^H NMR spectrum of the complex between **15** and methyl orange.

Analysis of the ^1^H NMR spectrum of pillar[5]arene **13** with **G2** showed insignificant changes, which did not allow the structure of the complex to be unambiguously established. For this reason two-dimensional ^1^H-^1^H NOESY and diffusion-ordered NMR spectroscopy (DOSY) were used to establishing through-space interactions of the complex. NMR spectra were recorded in DMSO-d_6_ at a 1:1 ratio of each component at 1 × 10^−2^ M.

A cross-peak between S-CSA H^c^ protons and H^2^ protons of the methyl fragment of macrocycle **13** was observed in the ^1^H-^1^H NOESY NMR spectrum (Figure 6). Cross-peaks between protons of the methyl group H^g^ of guest **G2** and H^2^ and H^5^ protons of pillar[5]arene **13** were also observed. The ability to correlate spectral parameters with self-diffusion coefficients makes DOSY an important experimental method as it allows the separation of the spectral contributions of the system’s components based on their size differences [31,46]. DOSY NMR spectra were recorded in DMSO-d_6_ at the 1 × 10^−2^ M. The formation of the **13**/**G2** complex was further confirmed by two-dimensional DOSY NMR spectroscopy (Appendix A) and diffusion coefficients of **13**, **G2** and **13**/**G2** were determined (Appendix A). The DOSY spectrum of **13**/**G2** mixture in a 1:1 ratio (Appendix A) indicates the presence of only one type of particles with a diffusion coefficient lower than macrocycle **13** and camphorsulfonic acid **G2**. An additional criterion confirming the formation of an associate between **13** and **G2** is a significant decrease in diffusion rate **13**/**G2** complex, which also indicates the formation of stable associates.

#### 2.4.3. Aggregation Study of pillar[5]arenes Containing Amino Acid Residues with Guests by DLS and TEM

Dynamic light scattering (DLS) is used to determine particles’ size [47,48]. Therefore, the next stage of investigation was the study of pillar[5]arenes’ **12**–**15** ability to form supramolecular assemblies in the presence of guest molecules **G1**-**G4** in both water and DMSO. No association was detected in systems formed from mixtures of macrocycles **12**–**15** and guests **G1**–**G4** in water. It also was shown that the compounds **12**, **14** and **15** do not form associates with guests **G1**–**G4** in DMSO. The formation of aggregates at the same time was found only for a solution of pillar[5]arene **13** (1 × 10^−5^ M) with S-CSA which had an average hydrodynamic diameter of 713 ± 40 nm (PDI = 0.29) at a 1:1 ratio (Appendix A). It is worth noting that macrocycle **13** forms associates only in aqueous solution. It can be hypothesized that for **13**/**G2** complex (or mixture) in DMSO (1S)-(+)-10-camphorsulfonic acid **G2** plays the role of the chiral template around which novel structures are formed. It is well known [49] that the nature of solvent has a significant influence on self-assembly of pillararenes. DMSO usually completely prevents self-association of macrocycles, the formation of supramolecular polymers and other possible types of supramolecular structures. In this regard, it can be expected that inclusion complexes upon the interaction of the pillar[5]arene with the guest in DMSO will be formed. However, experimental data have unambiguously shown that interaction occurs only in the case of aggregates formation between host **13** and guest **G2**. Thus, the interaction of macrocycles **12**–**15** with the studied guests **G1**-**G4** in DMSO is observed only for host-guest pair **13** and **G2**. This is due to the recognition of submicron aggregates during self-assembly. There is no association and the following aggregation in the other cases. The presence of asymmetric induction and the minimum substituent (Me) in the amino acid residue obviously turned out to be a decisive factor in the interaction of pillar[5]arene **13** with bulky camphorsulfonic acid **G2**. In the same time the complex formation between pillar[5]arene **15** with methyl orange dye is due to the presence of the most hydrophobic pseudocavity in macrocycle **15** in the series of synthesized compounds **12**–**15**, which is in good agreement with the previously presented works [42,43].

The presence of aggregates formed by inclusion complex of pillar[5]arene **13** and **G2** with submicron size was also confirmed by TEM. The Figure 7 clearly shows the presence of spherical particles formed due to the conglomeration of inclusion complexes in large aggregates. This is in good agreement with the data obtained by the DLS method, from which the average hydrodynamic diameter was 700 nm.

#### 2.4.4. Theoretical Chemical Calculations

The semi-empirical PM6 method was used to predict the structure of the complex formed by pillar[5]arene **13** and (1S)-(+)-10-camphorsulfonic acid **G2**. Conformational analysis was used to determine the conformer with the lowest steric energy (Figure 8). The Gibbs free energies of macrocycle **13**, camphorsulfonic acid and the complex between the two were calculated and gave a value for ∆G_(formation)_ of −54.58 kJ·mol^−1^. This indicates that complexation between the compound **13** and **G2** guest is favorable and an energy efficient process.

#### 2.4.5. CD Spectroscopy

The final stage of investigation was the study of the complex formed by pillar[5]arene **13** and guest **G2** by CD spectroscopy (Figure 9, Appendix A). Solutions of macrocycle **13** were prepared in DMSO and water at the 1 × 10^−4^ M concentration. The concentration of camphorsulfonic acid was also 1 × 10^−4^ M. It was shown (Figure 9) that the Cotton effect decreased at 300 nm for the **13**/**G2** mixture in DMSO, while for the solution of **13** with S-CSA in water it is slightly increased. Such changes in CD spectra of pillar[5]arene **13**–camphorsulfonic acid solutions are probably due to self-assembly into the corresponding aggregates.

## 3. Materials and Methods

### 3.1. General

^1^H NMR, ^13^C and 2D NOESY NMR spectra were obtained on a Bruker Avance-400 spectrometer (Bruker Corp., Billerica, MA, USA) (^13^C{^1^H}—100 MHz and ^1^H and 2D NOESY—400 MHz). The chemical shifts were determined against the signals of residual protons of deuterated solvent (CDCl_3_, CD_3_C(O)CD_3_, D_2_O, DMSO-d_6_). The concentrations of the compounds were equal to 3–5% in all the records. Spectrum 400 (PerkinElmer) IR spectrometer, Bruker Ultraflex III MALDI-TOF (p-nitroaniline matrix, electrospray ionization with positive ions registration in the m/z range from 100 to 2800) were applied for the IR and mass spectra recording, respectively, and PerkinElmer 2400 Series II for elemental analysis. DataAnalysis 4.0 software (Bruker Daltonik GmbH, Bremen, Germany) was used for spectra analysis. The circular dichroism was measured with the Jasco-1500 spectrophotometer in 1 mm thick quartz cuvettes (15 nm/min, 235–330 nm, slit width 1 nm, sampling step 1 nm, 3 scans co-addition). Boron tribromide, 1,4-dimethoxybenzene, and ethyl bromoacetate were purchased from Acros and used as received. All the aqueous solutions were prepared with the Millipore-Q deionized water (>18.0 MΩ cm at 25 °C).

**4,8,14,18,23,26,28,31,32,35-Deca[(ethoxycarbonyl)methoxy]-pillar[5]arene** (**10**) was prepared by a literature method [39].

**4,8,14,18,23,26,28,31,32,35-Deca-[(*N*-3′,3′-dimethylaminopropyl)-carbamoylmethoxy]-pillar[5]arene** (**11**) was prepared by a literature method [40].

***N*-bromoacetyl-glycine ethyl ester** (**3**), ***N*-bromoacetyl-glycylglycine ethyl ester** (**5**), ***N*-bromoacetyl-*L*-alanine ethyl ester** (**7**) and ***N*-bromoacetyl-*L*-phenylalanine** (**9**) were prepared by a literature methods [38].

***N*-[3-(dimethylamino)propyl]-2-(4-methoxyphenoxy)acetamide** (**16**) was prepared by a literature method [50].

### 3.2. General Procedure for the Synthesis of pillar[5]arenes **12**–**15**

In a round-bottom flask equipped with magnetic stirrer 0.1 g (0.05 mmol) of decaamine **11** was dissolved in 5 mL of acetonitrile, and 0.50 mmol of alkylating reagents were added. The reaction mixture was refluxed for 12–14 h. Solvent was removed under reduced pressure. The precipitates were dried under reduced pressure over P_2_O_5_.

**4,8,14,18,23,26,28,31,32,35-Deca-[(*N*-(3′,3′-dimethyl-3′-{(ethoxycarbonylmethyl)amidocarbonylmethyl}ammoniumpropyl)carbamoyl-methoxy]-pillar[5]arene decabromide** (**12**). Yield: 0.19 g (89%). M.P. = 82–83 °C. ^1^H NMR (DMSO-*d_6_*, 400 MHz, 298 K): *δ*_H_, ppm, *J*/Hz: 1.16 (t, 30H, CH_3_CH_2_O, ^3^*J*_H,H_ = 7.0 Hz), 1.97–1.99 (m, 20H, NHCH_2_CH_2_CH_2_N), 3.24–3.26 (m, 80H, NHCH_2_CH_2_CH_2_N and N(CH_3_)_2_), 3.54–3.60 (m, 30H, ArCH_2_Ar and NHCH_2_CH_2_CH_2_N), 3.92–3.99 (m, 20H, NHCH_2_COOEt), 4.04–4.10 (m, 20H, CH_3_CH_2_O), 4.20–4.24 (m, 20H, N(CH_3_)_2_CH_2_), 4.40–4.51 (m, 20H, ArOCH_2_), 6.84 (br.s, 10H, ArH), 8.20–8.31 (m, 10H, ArOCH_2_C(O)NH), 9.12 (br.t, 10H, C(O)NHCH). ^13^C NMR (DMSO-*d_6_*, 100 MHz, 298 K) *δ*_C_, ppm: 14.1, 22.8, 35.6, 40.8, 41.1, 51.3, 60.9, 61.8, 63.0, 67.6, 114.6, 127.9, 149.0, 163.9, 168.6, 169.2. IR(*ν*/cm^−1^): 1199 (C-O-C), 1675 (C=O), 2960 (N–H), 3331 (N–H). MS (ESI): calculated [M − 3Br^−^]^3+^
*m/z* = 1343.8, [M − 4Br^−^]^4+^
*m/z* = 987.6, [M − 5Br^−^]^5+^
*m/z* = 774.3, [M − 6Br^−^]^6+^
*m/z* = 632.1, [M − 7Br^−^]^7+^
*m/z* = 530.4, found [M − 3Br^−^]^3+^
*m/z* = 1344.6, [M − 4Br^−^]^4+^
*m/z* = 988.6, [M − 5Br^−^]^5+^
*m/z* = 774.7, [M − 6Br^−^]^6+^
*m/z* = 632.4, [M − 7Br^−^]^7+^
*m/z* = 530.5. El. anal. calcd for C_165_H_270_Br_10_N_30_O_50_: C 46.38, H 6.37, N 9.83, Br 18.70. Found: C 46.44, H 6.13, N 9.90, Br 18.64.

**4,8,14,18,23,26,28,31,32,35-Deca-[(*N*-(3′,3′-dimethyl-3′-{(ethoxycarbonyl[*S*-methyl]methyl)amidocarbonylmethyl}ammoniumpropyl)carbomoyl-methoxy]-pillar[5]arene decabromide** (**13**): Yield: 0.20 g (93%). M.P. = 68–70 °C. ^1^H NMR (DMSO-*d_6_*, 400 MHz, 298 K): *δ*_H_, ppm, *J*/Hz: 1.18 (t, 30H, CH_3_CH_2_O, ^3^*J*_H,H_ = 6.8 Hz), 1.33 (d, 30H, NHCHCH_3_, ^3^*J*_H,H_ = 6.4 Hz), 1.96–1.99 (m, 20H, NHCH_2_CH_2_CH_2_N), 3.18–3.27 (m, 80H, NHCH_2_CH_2_CH_2_N and N(CH_3_)_2_), 3.50–3.66 (m, 30H,ArCH_2_Ar and NHCH_2_CH_2_CH_2_N), 4.0–4.16 (m, 20H, OCH_2_CH_3_), 4.17–4.25 (m, 20H, N(CH_3_)_2_CH_2_), 4.26–4.31 (m, 10H, NHCH(CH_3_)COOEt), 4.33–4.52 (m, 20H, ArOCH_2_), 6.84 (br.s, 10H, ArH), 8.20–8.40 (m, 10H, ArOCH_2_C(O)NH), 9.19 (br.s, 10H, C(O)NHCH). ^13^C NMR (DMSO-*d_6_*, 100 MHz, 298 K) *δ*_C_, ppm: 14.0, 16.5, 22.7, 35.6, 48.1, 51.2, 60.8, 61.7, 63.1, 67.6, 114.6, 127.9, 149.0, 163.1, 168.5, 171.8. IR(*ν*/cm^−1^): 1157 (C-O-C), 1676 (C=O), 2961 (N–H), 3330 (N–H). MS (ESI): calculated [M − 3Br^−^]^3+^
*m/z* = 1390.5, [M − 4Br^−^]^4+^
*m/z* = 1022.7, [M − 5Br^−^]^5+^
*m/z* = 802.3, [M − 6Br^−^]^6+^
*m/z* = 655.5, [M − 7Br^−^]^7+^
*m/z* = 550.4, [M − 8Br^−^]^8+^
*m/z* = 471.6, found [M − 3Br^−^]^3+^
*m/z* = 1391.3, [M − 4Br^−^]^4+^
*m/z* = 1023.7, [M − 5Br^−^]^5+^
*m/z* = 802.9, [M − 6Br^−^]^6+^
*m/z* = 655.5, [M − 7Br^−^]^7+^
*m/z* = 550.5, [M − 8Br^−^]^8+^
*m/z* = 471.7. El. anal. calcd for C_175_H_290_Br_10_N_30_O_50_: C 47.63, H 6.62, N 9.52, Br 18.10. Found: C 46.98, H 6.78, N 9.83, Br 17.94.

**4,8,14,18,23,26,28,31,32,35-Deca[(*N*-(3′,3′-dimethyl-3′-{([ethoxycarbonylmethyl]amidocarbonylmethyl)amidocarbonylmethyl})ammoniumpropyl)carbomoyl-methoxy]-pillar[5]arene decabromide** (**14**): Yield: 0.21 g (90%). M.P. = 82–84 °C. ^1^H NMR (DMSO-*d_6_*, 400 MHz, 298 K): *δ*_H_, ppm, *J*/Hz: 1.17 (t, 30H, CH_3_CH_2_O, ^3^*J*_H,H_ = 6.8 Hz), 1.80–2.06 (m, 20H, NHCH_2_CH_2_CH_2_N), 3.17–3.27 (m, 80H, NHCH_2_CH_2_CH_2_N and N(CH_3_)_2_), 3.50–3.61 (m, 30H, ArCH_2_Ar and NHCH_2_CH_2_CH_2_N), 3.86 (br.d, 40H, NHCH_2_C(O)NHCH_2_ and NHCH_2_C(O)NHCH_2_), 4.02–4.15 (m, 20H, OCH_2_CH_3_), 4.17–4.25 (m, 20H, N(CH_3_)_2_CH_2_), 4.30–4.50 (m, 20H, ArOCH_2_), 6.83 (br.s, 10H, ArH), 8.24–8.31 (m, 10H, ArOCH_2_C(O)NH), 8.53 (br.t, 10H, NHCH_2_C(O)NHCH_2_C(O)), 8.93 (br.s, 10H, C(O)NHCH_2_). ^13^C NMR (DMSO-*d_6_*, 100 MHz, 298 K) *δ*_C_, ppm: 14.1, 22.8 35.7, 40.7, 41.7, 51.2, 60.6, 62.1, 63.1, 67.6, 114.6, 127.9, 149.1, 163.6, 168.6, 169.7. IR(*ν*/cm^−1^): 1197 (C-O-C), 1667 (C=O), 2964 (N–H), 3056 (N–H). MS (ESI): calculated [M − 6Br^−^ − CH_2_C(O)NHCH_2_C(O)OCH_2_CH_3_]^6+^
*m/z* = 703.5, found [M − 6Br^−^ − CH_2_C(O)NHCH_2_C(O)OCH_2_CH_3_]^6+^
*m/z* = 703.5. El. anal. calcd for C_185_H_300_Br_10_N_40_O_60_: C 45.87, H 6.24, N 11.57, Br 16.50. Found: C 46.02, H 6.09, N 10.82, Br 16.24.

**4,8,14,18,23,26,28,31,32,35-Deca-[(*N*-(3′,3′-dimethyl-3′-{(ethoxycarbonyl[*S*-benzyl] methyl)amidocarbonylmethyl}ammoniumpropyl)carbamoyl-methoxy]-pillar[5]arene decabromide** (**15**): Yield 0.22 g (87%). M.P. = 78–79 °C. ^1^H NMR (DMSO-*d_6_*, 400 MHz, 298 K): *δ*_H_, ppm, *J*/Hz: 1.11 (t, 30H, CH_3_CH_2_O, ^3^*J*_H,H_ = 7.0 Hz), 1.75–1.99 (m, 20H, NHCH_2_CH_2_CH_2_N), 2.81–3.01 (m, 20H, CHCH_2_Ph), 3.06–3.27 (m, 20H, NHCH_2_CH_2_CH_2_N), 3.13 (s, 60H, N(CH_3_)_2_), 3.45–3.65 (m, 30H, ArCH_2_Ar and NHCH_2_CH_2_CH_2_N), 4.01–4.23 (m, 10H, CHCH_2_Ph), 4.32–4.50 (m, 20H, ArOCH_2_), 4.50–4.60 (m, 20H, NCH_2_C(O)), 6.83 (br.s, 10H, ArH), 7.20–7.31 (m, 50H, Ar^Ph^H), 8.23–8.31 (m, 10H, ArOCH_2_C(O)NH), 9.20 (d, 10H, C(O)NHCH, ^3^*J*_H,H_ = 7.2 Hz). ^13^C NMR (DMSO-*d_6_*, 100 MHz, 298 K) *δ*_C_, ppm: 14.4, 23.1, 36.1, 37.0, 37.2, 51.1, 53.7, 61.0, 61.7, 63.0, 126.8, 128.3, 129.2, 136.6, 163.2, 168.5, 170.7. IR(*ν*/cm^−1^): (C-O-C), 1677 (C=O), 3191 (N-H). MS (ESI): calculated [M − 10Br^−^ + 2H^+^]^12+^
*m/z* = 365.9, found [M − 10Br^−^ + 2H^+^]^12+^
*m/z* = 364.5. El. anal. calcd for C_235_H_330_Br_10_N_30_O_50_: C 54.55, H 6.43, N 8.12, Br 15.44. Found: C 69.87, H 7.13, N 3.30, Br 15.62.

### 3.3. General Procedure for the Synthesis of Compounds **17** and **18**

In a round-bottom flask equipped with magnetic stirrer 0.1 g (0.38 mmol) of amine 16 was dissolved in 3 mL of acetonitrile, and 0.38 mmol of alkylating reagents were added. The reaction mixture was refluxed for 8 h. Solvent was removed under reduced pressure. The precipitates were dried under reduced pressure over P_2_O_5_.

***N*-(2-((1-Ethoxy-1-oxopropan-2-yl)amino)-2-oxoethyl)-3-(2-(4-methoxyphenoxy)acetamido)-*N*,*N*-dimethylpropan-1-ammonium bromide** (**17**): Yield: 0.17 g (95%). Yellow oil. ^1^H NMR (CDCl_3_, 400 MHz, 298 K): *δ*_H_, ppm, *J*/Hz: 1.23 (t, 3H, CH_3_CH_2_O, ^3^*J*_H,H_ = 7.8 Hz), 1.24 (d, 3H, NHCH(CH_3_)CO, ^3^*J*_H,H_ = 6.0 Hz), 2.14–2.27 (m, 2H, NHCH_2_CH_2_CH_2_N^+^), 3.37 (s, 3H, N(CH_3_)), 3.40 (s, 3H, N(CH_3_)), 3.46–3.51 (m, 2H, NHCH_2_CH_2_CH_2_N^+^), 3.61–3.74 (m, 2H, NHCH_2_CH_2_CH_2_N^+^), 3.75 (s, 3H, ArOCH_3_), 4.06–4.13 (m, 2H, C(O)OCH_2_CH_3_), 4.19 and 4.96 (d, 2H, N^+^CH_2_C(O), ^3^*J*_H,H_ = 13.6 Hz), 4.31–4.46 (m, 2H, NHCH_2_(CH_3_)), 4.47 (s, 2H, ArOCH_2_), 6.81–6.92 (m, 4H, ArH), 7.43 (t, 1H, ArOCH_2_C(O)NH, ^3^*J*_H,H_ = 6.0 Hz), 9.30 (d, 1H, C(O)NHCH, ^3^*J*_H,H_ = 6.4 Hz). ^13^C NMR (CDCl_3_, 100 MHz, 298 K) *δ*_C_, ppm: 14.2, 16.5, 23.5, 25.4, 35.9, 49.1, 52.4, 52.6,55.8, 61.6, 62.8, 64.4, 68.1, 114.9, 116.0, 151.5, 154.7, 162.8, 169.7, 172.1. IR(*ν*/cm^−1^): 1541 (C(O)NH, amide I), 1674 (C(O)NH, amide II), 3207 (N–H), 3331 (N–H). MS (ESI): calculated [M − Br^−^]^+^
*m/z* = 424.2, [M + Br^−^]^-^
*m/z* = 584.1, found [M − Br^−^]^+^
*m/z* = 424.2, [M + Br^−^]^-^
*m/z* = 584.1. El. anal. calcd for C_21_H_34_BrN_3_O_6_: C 50.00, H 6.79, N 8.33, Br 15.84. Found: C 50.27, H 6.13, N 8.75, Br 15.64.

***N*-(2-((1-Ethoxy-1-oxo-3-phenylpropan-2-yl)amino)-2-oxoethyl)-3-(2-(4-methoxyphenoxy)acetamido)-*N,N*-dimethylpropan-1-ammonium bromide** (**18**): Yield: 0.19 g (94%). Yellow oil. ^1^H NMR (CDCl_3_, 400 MHz, 298 K): *δ*_H_, ppm, *J*/Hz: 1.20 (t, 3H, CH_3_CH_2_O, ^3^*J*_H,H_ = 7.0 Hz), 2.02–2.14 (m, 2H, NHCH_2_CH_2_CH_2_N^+^), 3.09–3.29 (m, 2H, CH(CH_2_Ar^Ph^)), 3.16 (s, 3H, N(CH_3_)), 3.23 (s, 3H, N(CH_3_)), 3.40–3.49 (m, 2H, NHCH_2_CH_2_CH_2_N^+^), 3.51–3.70 (m, 2H, NHCH_2_CH_2_CH_2_N^+^), 3.75 (s, 3H, ArOCH_3_), 4.04–4.13 (m, 2H, OCH_2_CH_3_), 4.24–4.30 and 4.75–4.80 (m, 2H, N^+^CH_2_C(O)), 4.46 (s, 2H, ArOCH_2_), 4.60–4.71 (m, 1H, CH(CH_2_Ar^Ph^)), 6.81–6.92 (m, 4H, ArH), 7.17–7.41 (m, 5H, Ar^Ph^H), 7.32 (t, 1H, ArOCH_2_C(O)NH, ^3^*J*_H,H_ = 6.0 Hz), 9.39 (d, 1H, C(O)NHCH, ^3^*J*_H,H_ = 7.6 Hz). ^13^C NMR (CDCl_3_, 100 MHz, 298 K) *δ*_C_, ppm: 14.2, 23.5, 35.9, 37.0, 52.3, 54.8, 55.8, 61.8, 62.6, 64.5, 68.1, 114.9, 116.0, 127.0, 128.6, 129.6, 136.8, 151.5, 154.7, 163.0, 169.7, 171.1. IR(*ν*/cm^−1^): 1542 (C(O)NH, amide I), 1674 (C(O)NH, amide II), 3032 (N–H), 3198 (N–H). MS (ESI): calculated [M − Br^−^]^+^
*m/z* = 500.3, found [M − Br^−^]^+^
*m/z* = 500.3. El. anal. calcd for C_21_H_34_BrN_3_O_6_: C 55.86, H 6.60, N 7.24, Br 13.76. Found: C 55.93, H 6.78, N 7.58, Br 13.84.

### 3.4. UV-Spectroscopy

Absorption spectra were recorded on a Shimadzu UV-3600 spectrometer (Kyoto, Japan). Quartz cuvettes with an optical path length of 10 mm were used. DMSO and water were used for preparation of the solutions. Absorption spectra of mixtures were recorded after an 1 h incubation at 20 °C. Solutions of **G1**-**G3** guests (*p*-toluenesulfonic acid, (1S)-(+)-10-camphorsulfonic acid, camphor) were added to those of compounds **12**–**15** (3 × 10^−5^ M) in a 1:10 ratio to study the complexation of pillar[5]arenes with guests in solvent. A 1:1 ratio was chosen in the case of methyl orange dye **G4**.

#### 3.4.1. Determination of the Stability Constant and Stoichiometry of the Complex by Spectrophotometric Titration

A 3 × 10^–3^ M solution of **G2** (30, 60, 90, 120, 150, 180, 210, 240, 270 and 300 μL) in DMSO was added to 0.3 mL of a solution of **13** (3 × 10^−4^ M) in DMSO and diluted to final volume of 3 mL with DMSO. The UV spectra of the solutions were then recorded. The stability constant of complex was calculated by Bindfit. Three independent experiments were carried out for each series.

3 × 10^–5^ M solution of **15** (300, 500, 600, 800, 900 1000, 1100, 1300, 1500 and 2000 μL) in water was added to 0.03 mL of the solution of **G4** (1 × 10^−3^ M) in water and diluted to final volume of 3 mL with water. The UV spectra of the solutions were then recorded. The stability constant of each complex was calculated by Bindfit. Three independent experiments were carried out for each series.

#### 3.4.2. Job Plots

Job Plots for hosts **13** and **15** and guests **G4** and **G2** were determined in deionized water and DMSO with the ratio varied from 0.6:2.4 to 2.4:0.6. Each measurements of the series were performed in three times.

### 3.5. Computational Methods

For the prediction of the complex structure was used the Spartan ’18 Parallel Suite [51]. The energy of complex formation was calculated based on the assumption that: ∆G_(formation)_ = ∆G_(complex)_ − [∆G_(host)_ + ∆G_(guest)_].

### 3.6. Dynamic Light Scattering (DLS)

The Zetasizer Nano ZS instrument (Worcestershire, UK) equipped with the 4 mW He-Ne laser (633 nm) was used for the determination of particle size. The aggregation study was carried out at 1:1 host-guest ratio. The concentration range for the pillar[5]arenes **12**–**15** was 1 × 10^−3^-1 × 10^−5^ M. Measurements were determined in 24 and 178 h for three times for the evaluation kinetic stability.

### 3.7. Transmission Electron Microscopy (TEM)

TEM analysis was carried out according to the literature [20] with the Hitachi HT7700 Exalens microscope (Tokyo, Japan).

### 3.8. ^1^H diffusion Ordered Spectroscopy (DOSY)

The DOSY NMR spectra were recorded on a Bruker Avance 400 spectrometer (Bruker Corp., Billerica, MA, USA) according to the literature [34,46].

## 4. Conclusions

The effect of the amino acid residues in the **12**–**15** series on the complexing and aggregation properties of water-soluble pillar[5]arenes was established. It was shown that the pillar[5]arene containing *L*-Ala fragments **13** bound (1S)-(+)-10-camphorsulfonic acid **G2** with an association constant of 613 M^−1^. The introduction of *L*-Phe fragments into the structure led to binding of methyl orange dye **G4** with an association constant of 4523 M^−1^. It was shown that particles formed by pillararenes containing Gly (**12**) and *L*-Ala (**14**) fragments had average hydrodynamic diameters of 165 and 238 nm, respectively. It was found for the first time that (1S)-(+)-10-camphorsulfonic acid can template the assembly of chiral particles resulting in **13** particles formation with a hydrodynamic diameter of 713 nm. The formation of supramolecular architectures with submicron size by **13** and **G2** was confirmed by TEM. The new generation of peptidomimetics based on decasubstituted water-soluble pillar[5]arenes have the potential in creation of novel stereoselective catalysts and transmembrane amino acid channels.

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
