# Peer review of "Self-Assembly of Supramolecular Architectures by the Effect of Amino Acid Residues of Quaternary Ammonium Pillar[5]arenes"

_ijms, 2020, doi:10.3390/ijms21197206_

Round 1

Reviewer 1 Report

The authors report the synthesis, characterization, complexation studies with specific guests and a study of the aggregation behavior of a series of pillar[5]arenes functionalized with aminoacids on their rims. The experimental work is complete with full characterization of the derivatives synthesized, detailed spectra analysis, detailed binding studies with the aid of NMR spectroscopy and photophysical measurements and complementary theoretical calculations. The presented chemistry has afforded worthy knowledge on the binding behavior of the functionalized pillar[5]arenes with two specific guests and the formation of their templated supramolecular structures, investigated by TEM. Furthermore, the paper is well supported by the references.

In my opinion, the paper deserves publication in International Journal of Molecular Sciences after the following revision/corrections have been made by the authors.

  • In Scheme 1, the molecules are too congested and thus, the authors should re-draw it in a way that it will be easy for the reader to follow it.
  • There are some mistakes in the manuscript that the authors should correct. For example, in page 4, line 107, the H14 is not a methylene proton but a methyl proton.
  • In the experimental section, the melting points of the solid products are mentioned as Dp. They should be reported as m.p. In addition, I would expect that the m.p. should have a range as the products are not recrystallized. The authors should correct them.
  • Finally, the language of the manuscript should be improved.

Author Response

  1. In Scheme 1, the molecules are too congested and thus, the authors should re-draw it in a way that it will be easy for the reader to follow it.

The Scheme 1 has been re-drawn.

  1. There are some mistakes in the manuscript that the authors should correct. For example, in page 4, line 107, the H14 is not a methylene proton but a methyl proton.

In page 4, line 107, “the methylene protons” has been corrected to “methyl protons”.

  1. In the experimental section, the melting points of the solid products are mentioned as Dp. They should be reported as m.p. In addition, I would expect that the m.p. should have a range as the products are not recrystallized. The authors should correct them.

“Dp” in the experimental section has been replaced by “M.p.” throughout the experimental section. The melting point values have also been corrected.

  1. Finally, the language of the manuscript should be improved.

The language of the manuscript has been improved.

Reviewer 2 Report

The submitted paper consist preparation of macromolecular structures based on pillar[5]arene. This is good documented synthetic work with detailed analysis of results supported by semi-empirical calculations.
In general this is good paper, however needs to be improved and some minor changes are necessary, as follows:
- lines 14-15 - change "Novel water-soluble multifunctional pillar[5]arenes containing amide, ammonium and amino acid fragments were synthesized" into "Novel water-soluble multifunctional pillar[5]arenes containing amide-ammonium-amino acid moiety were synthesized"; oryginal sentence indicate that it can be separately substituted by mentioned fragments which is not a case.
- line 25 - in Keywords the term "amino acid" is missing; this is an important point in the paper
- lines 56-57 - again this suggests that it is separately substituted by mentioned fragments which is not a case change into “containing amide-ammonium-amino acid moiety” or similar
- lines 58-59 - in the phrase "with a number of optically active" where is this 'number', there is only one (1S)-(+)-10-camphorsulfonic acid or maybe reference is missing
- lines 65-66 - change the position of ref. [33] for "the Ogoshi group [33]."
- lines 75-79 - rewrite "In order to obtain quaternary ammonium salts based on pillar[5]arene containing amino acid residues the highly reactive reagents (Scheme 1) were prepared according to the literature [38]." In reference 33 there is no original procedure, add appropriate reference or procedure in experimental. In this sentence numbers of compounds are omitted as depicted on Scheme 1; is to general without clear description, e.g., on the Scheme 1 for preparation hydrochloride 2 the thionyl chloride is used which is very unusual and have to be explain why? For the first look the Scheme 1 is too crowded, please change. Also the caption for this Scheme "Synthetic route for the preparation of alkylating reagents 3, 5, 7, 9." should be changed into "Synthetic route for preparation of alkyl bromides" this is a point in this synthesis. (By the way almost the same Scheme looks better in ref. 33)
- line 84 - change “β-phenylalanine” into “L-phenylalanine”
- line 91 - change “obtained with high 87-93% yields” into “obtained in high 87-93% yields”
- lines 101-102 - change “Cotton effect decreases with the 101 decrease of the enantiomeric purity” for more elegant “Cotton effect diminish with the decrease of the enantiomeric purity”
- line 138 - add on the Scheme 3 “reflux” under CH3CN
- line 142 - change “in acetonitrile” into „in boiling acetonitrile”
- line 163 - change “4-methoxyphenol derivatives, 17 and 18,” into “ derivatives 17 and 18”
- lines 198-199 - change “p-Toluenesulfonic acid, 198 (1S)-(+)-10-camphorsulfonic acid, camphor and methyl orange dye” into “p-Toluenesulfonic acid G1, (1S)-(+)-10-camphorsulfonic acid G2, camphor G3 and methyl orange dye G4”
- lines 201-202 - remove “(p-toluenesulfonic 201 acid, (1S)-(+)-10-camphorsulfonic acid, camphor)”
- line 204 - remove “methyl orange dye”
- line 235 – change “camphorsulfonic acid and methyl orange dye respectively” into “G2 and G4, respectively”
- line 227 - change “(1S)-(+)-10-camphorsulfonic acid” into “G2”
- line 241 - change “camphorsulfonic acid” into “G2”
- line 280 - change “for 13/G2 in DMSO (1S)-(+)-10-camphorsulfonic acid” into “for 13/G2 complex (or mixture) in DMSO (1S)-(+)-10-camphorsulfonic acid G2”
- line 297 - remove “S-camphorsulfonic acid”
- line 303 - “2.4.4. Theoretical chemical calculations” too small font
- line 506 – change “in the Gly – Ala – GlyGly – Phe series” into “in the 12-15 series”
- line 508 - change “pillar[5]arene containing L-Ala fragments bound (1S)-(+)-10-camphorsulfonic acid” into “pillar[5]arene containing L-Ala fragments 13 bound (1S)-(+)-10-camphorsulfonic acid G2”
- line 510 - change “methyl orange dye” into “methyl orange dye G4”
- line 511 – change “pillararenes containing Gly and L-Ala fragments had” into “pillararenes containing Gly and L-Ala fragments 12-14 had”
- line 513 - change “pillar[5]arene with L-Ala fragments” into “13”
- line 514 - change “pillar[5]arene containing L-Ala residues and camphorsulfonic acid” into “13 and G2”
- line 534 – in my opinion, all Abbreviations should be removed; there are common and well known

Author Response

  1. In general this is good paper, however needs to be improved and some minor changes are necessary, as follows:

- lines 14-15 - change "Novel water-soluble multifunctional pillar[5]arenes containing amide, ammonium and amino acid fragments were synthesized" into "Novel water-soluble multifunctional pillar[5]arenes containing amide-ammonium-amino acid moiety were synthesized"; oryginal sentence indicate that it can be separately substituted by mentioned fragments which is not a case.

The sentence "Novel water-soluble multifunctional pillar[5]arenes containing amide, ammonium and amino acid fragments were synthesized" has been corrected to "Novel water-soluble multifunctional pillar[5]arenes containing amide-ammonium-amino acid moiety were synthesized".

  1. - line 25 - in Keywords the term "amino acid" is missing; this is an important point in the paper

The term “amino acid” has been added in Keywords.

  1. - lines 56-57 - again this suggests that it is separately substituted by mentioned fragments which is not a case change into “containing amide-ammonium-amino acid moiety” or similar

“amide, ammonium, and amino acid residues” has been changed into “amide-ammonium-amino acid moiety”.

  1. - lines 58-59 - in the phrase "with a number of optically active" where is this 'number', there is only one (1S)-(+)-10-camphorsulfonic acid or maybe reference is missing

p-Toluenesulfonic acid, (1S)-(+)-10-camphorsulfonic acid, camphor and methyl orange dye were chosen as substrates. Thus, there are two optically active guests, namely, p-toluenesulfonic acid and camphor.

  1. - lines 65-66 - change the position of ref. [33] for "the Ogoshi group [33]."

The position of ref. [33] has been changed.

  1. - lines 75-79 - rewrite "In order to obtain quaternary ammonium salts based on pillar[5]arene containing amino acid residues the highly reactive reagents (Scheme 1) were prepared according to the literature [38]." In reference 33 there is no original procedure, add appropriate reference or procedure in experimental. In this sentence numbers of compounds are omitted as depicted on Scheme 1; is to general without clear description, e.g., on the Scheme 1 for preparation hydrochloride 2 the thionyl chloride is used which is very unusual and have to be explain why? For the first look the Scheme 1 is too crowded, please change. Also the caption for this Scheme "Synthetic route for the preparation of alkylating reagents 3, 5, 7, 9." should be changed into "Synthetic route for preparation of alkyl bromides" this is a point in this synthesis. (By the way almost the same Scheme looks better in ref. 33).

The sentence "In order to obtain quaternary ammonium salts based on pillar[5]arene containing amino acid residues the highly reactive reagents (Scheme 1) were prepared according to the literature [38]." was rewritten according to reviewer comments.

The reference 38 has been changed.

The preparation of hydrochloride 2 using thionyl chloride on the Scheme 1 is unfortunate misprint, cause the compound 1 is not ether of amino acid. The compound 1 is amino acid - glycine.

The Scheme 1 has been redrawn.

The caption for the Scheme 1 has been changed into "Synthetic route for preparation of alkyl bromides".

  1. - line 84 - change “β-phenylalanine” into “L-phenylalanine”

“β-phenylalanine” has been corrected to “L-phenylalanine”.

  1. - line 91 - change “obtained with high 87-93% yields” into “obtained in high 87-93% yields”

“obtained with high 87-93% yields” has been changed to “obtained in high 87-93% yields”.

  1. - lines 101-102 - change “Cotton effect decreases with the 101 decrease of the enantiomeric purity” for more elegant “Cotton effect diminish with the decrease of the enantiomeric purity”

“Cotton effect decreases with the 101 decrease of the enantiomeric purity” has been corrected to “Cotton effect diminish with the decrease of the enantiomeric purity”.

  1. - line 138 - add on the Scheme 3 “reflux” under CH3CN

“reflux” has been added on the Scheme 3.

  1. - line 142 - change “in acetonitrile” into „in boiling acetonitrile”

“in acetonitrile” has been changed into „in boiling acetonitrile”.

  1. - line 163 - change “4-methoxyphenol derivatives, 17 and 18,” into “ derivatives 17 and 18”

“4-methoxyphenol derivatives, 17 and 18,” has been corrected to “derivatives 17 and 18”.

  1. - lines 198-199 - change “p-Toluenesulfonic acid, 198 (1S)-(+)-10-camphorsulfonic acid, camphor and methyl orange dye” into “p-Toluenesulfonic acid G1, (1S)-(+)-10-camphorsulfonic acid G2, camphor G3 and methyl orange dye G4”

“p-Toluenesulfonic acid, 198 (1S)-(+)-10-camphorsulfonic acid, camphor and methyl orange dye” has been changed into “p-Toluenesulfonic acid G1, (1S)-(+)-10-camphorsulfonic acid G2, camphor G3 and methyl orange dye G4”.

  1. - lines 201-202 - remove “(p-toluenesulfonic 201 acid, (1S)-(+)-10-camphorsulfonic acid, camphor)”

“(p-toluenesulfonic 201 acid, (1S)-(+)-10-camphorsulfonic acid, camphor)” has been removed.

  1. - line 204 - remove “methyl orange dye”

“methyl orange dye” has been removed.

  1. - line 235 – change “camphorsulfonic acid and methyl orange dye respectively” into “G2 and G4, respectively”

“camphorsulfonic acid and methyl orange dye respectively” has been corrected to “G2 and G4, respectively”.

  1. - line 227 - change “(1S)-(+)-10-camphorsulfonic acid” into “G2”

“(1S)-(+)-10-camphorsulfonic acid” has been changed into “G2”.

  1. - line 241 - change “camphorsulfonic acid” into “G2”

“camphorsulfonic acid” has been corrected into “G2”.

  1. - line 280 - change “for 13/G2 in DMSO (1S)-(+)-10-camphorsulfonic acid” into “for 13/G2 complex (or mixture) in DMSO (1S)-(+)-10-camphorsulfonic acid G2”

“for 13/G2 in DMSO (1S)-(+)-10-camphorsulfonic acid” has been changed into “for 13/G2 complex (or mixture) in DMSO (1S)-(+)-10-camphorsulfonic acid G2”.

  1. - line 297 - remove “S-camphorsulfonic acid”

“S-camphorsulfonic acid” has been removed.

  1. - line 303 - “2.4.4. Theoretical chemical calculations” too small font.

The font of “2.4.4. Theoretical chemical calculations” has been increased.

  1. - line 506 – change “in the Gly – Ala – GlyGly – Phe series” into “in the 12-15 series”

“in the Gly – Ala – GlyGly – Phe series” has been changed into “in the 12-15 series”.

  1. - line 508 - change “pillar[5]arene containing L-Ala fragments bound (1S)-(+)-10-camphorsulfonic acid” into “pillar[5]arene containing L-Ala fragments 13 bound (1S)-(+)-10-camphorsulfonic acid G2”

“pillar[5]arene containing L-Ala fragments bound (1S)-(+)-10-camphorsulfonic acid” has been changed into “pillar[5]arene containing L-Ala fragments 13 bound (1S)-(+)-10-camphorsulfonic acid G2”.

  1. - line 510 - change “methyl orange dye” into “methyl orange dye G4”

“methyl orange dye” has been changed into “methyl orange dye G4”.

  1. - line 511 – change “pillararenes containing Gly and L-Ala fragments had” into “pillararenes containing Gly and L-Ala fragments 12-14 had”

“pillararenes containing Gly and L-Ala fragments had” has been corrected to “pillararenes containing Gly (12) and L-Ala (14) fragments 12-14 had”.

  1. - line 513 - change “pillar[5]arene with L-Ala fragments” into “13”

“pillar[5]arene with L-Ala fragments” has been changed into “13”.

  1. - line 514 - change “pillar[5]arene containing L-Ala residues and camphorsulfonic acid” into “13 and G2”

“pillar[5]arene containing L-Ala residues and camphorsulfonic acid” has been changed into “13 and G2”.

  1. - line 534 – in my opinion, all Abbreviations should be removed; there are common and well known

All abbreviations have been removed.
